# The Antioxidant and Antihyperglycemic Activities of Bottlebrush Plant (*Callistemon lanceolatus*) Stem Extracts

**DOI:** 10.3390/medicines7030011

**Published:** 2020-03-04

**Authors:** Ramesh Kumar, Ashutosh Gupta, Amit Kumar Singh, Anupam Bishayee, Abhay K. Pandey

**Affiliations:** 1Department of Biochemistry, University of Allahabad, Allahabad 211 002, Uttar Pradesh, India; rameshbiochem91@gmail.com (R.K.); ashutosh8998@gmail.com (A.G.); amitfbs21@gmail.com (A.K.S.); 2Lake Erie College of Osteopathic Medicine, Bradenton, FL 34211, USA

**Keywords:** *Callistemon lanceolatus*, antidiabetic, antioxidant, hyperglycemia, serum markers, rats

## Abstract

**Background:** Diabetes mellitus, a metabolic disease, is a major health concern today throughout the world. *Callistemon lanceolatus* (Myrtaceae), commonly known as bottlebrush, has been used by Indian tribal communities for the treatment of many diseases. The purpose of this study was to explore antioxidant and antihyperglycemic potential of methanolic and aqueous extracts of the stem of C. lanceolatus in vitro and in vivo. **Methods:** Phytoconstituents of C. lanceolatus stem were extracted in methanol and water sequentially followed by phytochemical analysis. The in vitro antioxidant potential of aqueous and methanolic extracts was assessed by metal ion chelating, free radical scavenging, and reducing power assays. The in vivo antihyperglycemic activity of the oral methanolic extract was studied in alloxan-induced diabetic rats. Bodyweight and blood glucose were monitored regularly. After the treatment period, serum was examined for total cholesterol, triglycerides, high-density lipoprotein (HDL), bilirubin, creatinine, urea, glutamate pyruvate transaminase (SGPT), glutamate oxaloacetate transaminase (SGOT), and alkaline phosphatase (ALP). **Results:** Methanolic extract exhibited superior antioxidant activity to aqueous extract. A marked increase in levels of serum markers, viz., glucose, triglycerides, total cholesterol, bilirubin, urea, creatinine, SGOT, SGPT, and ALP along with a reduction in HDL was observed in diabetic rats. Methanol extract treatment for 28 days accounted for a decrease in blood glucose and other metabolic markers accompanied by an improvement in body weight and HDL level in hyperglycemic rats. **Conclusions:** The present study suggests that *C. lanceolatus* methanolic stem extract possesses antioxidant and antihyperglycemic activities and has potential as a therapeutic agent in diabetes.

## 1. Introduction

Free radicals or reactive oxygen species (ROS) are fundamental to biochemical processes and play a significant role in aerobic life and metabolism. They are produced inside the body via normal physiological processes as well as during detoxification of xenobiotics [1]. The excessive load of ROS in the living system might be associated with various pathological conditions. ROS attack biomolecules and produce structural alterations. Lipid peroxidation resulting from ROS action leads to decreased membrane fluidity and ultimately causes various diseases [2].

Diabetes mellitus (DM) is a chronic metabolic disorder represented by hyperglycemia, resulting from insufficient insulin secretion or its ineffective action with variations in carbohydrate, lipid, and protein metabolism. Findings from recent studies have indicated that hyperglycemia may cause the non-enzymatic glycosylation of numerous macromolecules along with ROS production and the alteration of endogenous antioxidant status that finally results in the progression of DM-associated prolonged complications [3].

Presently, India has the second largest diabetic population in the world. According to a report from the International Diabetes Federation published in 2017, India has currently about 72.9 million diabetic individuals, and it is expected to increase further up to 134.3 million in 2045 [4]. India recorded continuous rise in the occurrence of DM from 2% in the 1970s to over 20% at present and larger cities have witnessed more cases, and the rural areas are also being affected [4]. DM is a progressive metabolic disorder affecting a substantial percentage of the human population around the world. According to the WHO report, diabetes is spreading like an epidemic throughout the world. Areas under greater risk include the Western Pacific and Southeast Asia, and the major population have type-2 diabetes mellitus (T2DM) [5]. Free radical/ROS load is one of the important reasons for the development of T2DM, which is characterized by insulin resistance, imbalanced glycemic homeostasis, and progression towards DM-associated complications. Experimentally, diabetes is induced in rodents by injecting alloxan which accumulates in pancreatic β-cells and undergoes reduction to produce diluric acid and ROS, which subsequently causes the destruction of β-cells and the induction of diabetes [6]. Prolonged hyperglycemia induces oxidative stress via various interconnected metabolic pathways, producing perturbations in the endogenous antioxidant defence system, which is eventually accountable for the advancement of complications at micro/macro level [7,8]. According to the WHO report, about 80% of the people around globe still depend on the indigenous system of medicine, i.e., plant-based natural products for their primary healthcare needs [8,9]. Antioxidants are endogenously formed in the body and/or acquired through food supplements that maintain the equilibrium between generation and quenching of free radicals. Phytochemicals, such as alkaloids, phenolics, terpenes, and flavonoids, act as antioxidants to scavenge or diminish the generation of ROS [9,10].

*Callistemon lanceolatus (*Syn. *Callistemon citrinus*), a plant of the Myrtaceae family, is popularly known as bottlebrush, lemon bottlebrush, red bottlebrush, or crimson bottlebrush. It is a shrub or small tree that grows up to 7.5 m in height and produces masses of bright red bloom clusters that resemble a bottlebrush (Figure 1). This plant is native to Australia and has been introduced in India from Australia for ornamental purposes [10,11]. It is also commonly grown in other world regions, such as the United States (particularly in southern Florida and parts of southern California). Various parts of this plant have shown numerous biological and pharmacological activities, including antioxidant, anti-inflammatory antithrombin, elastase-inhibitory, antibacterial, antifungal, molluscicidal, neuroprotective, hepatoprotective, cardioprotective and anticancer activities [11,12,13,14,15,16,17,18]. However, reports regarding the antihyperglycemic activity of this plant are very scanty [19,20]. Moreover, unlike other parts, the stem of the plant has not been explored for its therapeutic potential. Hence, the current work was aimed to investigate the antioxidant and antihyperglycemic activities of *C. lanceolatus* stem extracts by using several in vitro and in vivo experimental approaches.

## 2. Materials and Methods 

### 2.1. Plant Material 

The *C. lanceolatus* plant sample was obtained from the Science Faculty Campus, University of Allahabad (Allahabad, India) in April 2018, and the plant identification was performed by Prof. Devendra K. Chauhan from the Department of Botany, University of Allahabad. The stem was surface-sterilized and shade-dried. The drying method involved spreading fresh plant material in a single layer and placing the materials outside under an awning where they are protected from direct sunlight. 

### 2.2. Preparation of Extract 

The shade-dried stem was ground using an electric grinder, and a powdered sample (particle size 0.7–1.0 mm) was serially extracted with methanol and water using Soxhlet apparatus for 8 h [7]. The temperature for extract preparation in methanol and water was 65 °C and 100 °C, respectively. The solvent was totally evaporated under reduced pressure. The dried extract was constituted in pure dimethyl sulfoxide (DMSO) for in vitro assays, whereas for in vivo assay it was constituted in water.

### 2.3. Phytochemical Screening

Different phytoconstituents, such as anthraquinone, flavonoids, saponins, tannins, alkaloids, phlobatannins, reducing sugars, terpenoids and cardiac glycosides, which are present in methanol as well as aqueous extracts were identified using standard chemical procedures [2,7].

### 2.4. Determination of Total Flavonoid Content 

The aluminum chloride colorimetric method [21], as previously modified by us [2], was used for the estimation of flavonoids in both the extracts. A little amount (0.2 mL) of extract constituted in DMSO (2 mg/mL) was mixed with methanol (1.8 mL), 10% aluminum chloride (0.1 mL), 1 M potassium acetate (0.1 mL), and distilled water (2.8 mL). The reaction mixture was incubated at 25 °C for 30 min, and absorbance was recorded at 415 nm using a spectrophotometer (Evolution 201, Thermo Scientific, Waltham, MA, USA). Quercetin was used as standard for preparing the calibration curve. The flavonoid content in the test sample was expressed as μg quercetin equivalent/mg sample (μg QE/mg). 

### 2.5. Determination of Total Phenolics

The total phenolic content in the extracts was determined according to the standard methods [21]. Sample (0.2 mL) was raised to 3 mL with water followed by the addition of two-fold-diluted Folin-Ciocalteau reagent (0.5 mL). After 3 min, 20% sodium carbonate solution (2 mL) was added and the tubes were heated in a boiling water bath for 1 min followed by cooling at room temperature. The absorbance was determined at 650 nm against a reagent blank using a spectrophotometer (Evolution 201, Thermo Scientific, Waltham, MA, USA). The concentration of phenolics in the test sample was expressed as µg propyl gallate equivalents/mg (µg PGE/mg). 

### 2.6. Free Radical Scavenging Assay

The free radical scavenging activity of the methanolic and aqueous extracts was determined in vitro using 1,1-diphenyl-2-picrylhydrazyl (DPPH) assay [22,23]. For this assay, extract was dissolved in DMSO instead of methanol. DPPH solution (3 mL, 0.1 mM) prepared in methanol was added to 1 mL of the test extracts (40–100 μg/mL). A similar concentration of butylated hydroxyl anisole (BHA) was used as a standard antioxidant compound for comparison. The reaction mixture was kept at room temperature for 30 min in dark followed by measurement of absorbance at 517 nm to observe the reduction in the DPPH free radical. A DPPH-free radical solution was used as a control. The percentage of scavenging activities (% inhibition) at different concentrations of the extract fractions was calculated using the following formula: $$\%\;\mathrm{Radical}\;\mathrm{scavenging}\;\mathrm{activity}\;=\frac{\left(\mathrm{Ac}-\mathrm{As}\right)}{\mathrm{Ac}}\times 100$$
where A_C_ and A_S_ represent absorbance of the control and the sample, respectively.

### 2.7. Reducing Power Assay

The reducing power of test extracts was measured by the method of Oyaizu [24] with minor changes [25]. To 1 mL aliquot of extracts (200–1000 μg/mL) prepared in DMSO, 2.5 mL of phosphate buffer (0.2 M, pH 6.6) and 2.5 mL of 1% potassium hexacyanoferrate [K_3_Fe(CN)_6_] were added. The tubes were incubated at 50 °C in a water bath for 20 min. The reaction was terminated by the addition of 10% trichloroacetic acid (2.5 mL) followed by centrifugation (4000× *g*, 10 min). The supernatant (1 mL) was mixed with distilled water (1 mL) and FeCl_3_ solution (0.5 mL, 0.1% *w*/*v*) and was incubated at 25 °C for 2 min. The absorbance was recorded at 700 nm. Ascorbic acid was used as a positive control for comparison. 

### 2.8. Metal Ion Chelating Activity

The ferrous ion chelating ability of *C. lanceolatus* stem extracts was measured by the method of Dinis et al. [26]. The extract (200 μL) at different concentrations (200–400 µg/mL) was added to 0.05 mL of ferrous chloride solution (2 mM). Propyl gallate, a standard antioxidant compound, was used as positive control for comparison. The reaction was initiated by adding ferrozine (0.2 mL, 5 mM). The mixture was shaken forcefully and left at room temperature for 10 min. The absorbance was measured at 562 nm. A ferrous chloride–ferrozine reaction mixture was used as a control. The percentage inhibition of ferrozine-Fe_2_^+^ complex formation was calculated by the formula given below:

Metal ion chelating ability (% inhibition of ferrozine-Fe_2_^+^complex formation)
$$\frac{\left(A0\;-A1\right)}{A0}\times 100$$
where *A*0 is the absorbance of the control and *A*1 is absorbance of the sample.

### 2.9. Animal and Maintenance 

Albino Wistar rats of about same age (weight 220–280 g) and of either sex were obtained from the Indian Institute of Toxicological Research (Lucknow, Uttar Pradesh, India). The rats were maintained at the departmental animal house (23 ± 2 °C) with 12-h light and dark cycles for 1 week before and during experiments. Animals were fed with a standard pellet diet (Paramaount Techno-Chem, Varanasi, Uttar Pradesh, India) and water was given ad libitum. The in vivo study was performed in accordance with the guidelines of the Institutional Animal Ethics Committee as approved by the National Committee for the Purpose of Control and Supervision of Experiments on Animals Ethical approval code: IAEC/AU/2019(1)/15; Date of approval: 1 February 2019. 

### 2.10. Experimental Protocol 

After the acclimatization period of 1 week, the rats were randomly divided into six groups, containing five animals each as follows:Group 1: Normal controlGroup 2: Diabetic controlGroup 3: Diabetic rats received reference drug glibenclamide (5 mg/kg) orally once a day for 4 weeks.Group 4: Diabetic rats received *C. lanceolatus* methanolic extract (200 mg/kg) orally once a day for 4 weeks.Group 5: Diabetic rats received *C. lanceolatus* methanolic extract (400 mg/kg) orally once a day for 4 weeks.Group 6: Extract control rats received *C. lanceolatus* methanolic extract daily (400 mg/kg) orally once a day for 4 weeks.

The alloxan monohydrate (Sisco Research Laboratories Private Limited, Mumbai, India) constituted in sterile normal saline was injected intraperitoneally (i.p.) in overnight-fasted rats at a dose of 80 mg/kg to induce diabetes. After 1 week, the induction of diabetes was confirmed in rats by the determination of fasting blood glucose (270–350 mg/dL) with the help of a glucometer (Accu-Check, Roche Diabetes Care GmbH Sandhofer Strasse Mannheim, Mallaustr, Germany). The rats were divided into various groups as mentioned earlier. Diabetic rats were treated with oral glibenclamide (5 mg/kg) or *C. lanceolatus* methanolic extract (200 or 400 mg/kg) for 4 weeks.

### 2.11. Biochemical Analysis

Blood samples were drawn from the tail vein of rats at weekly intervals until the end of the study. Body weight and fasting blood glucose measurements were done on day 1, 7, 14, 21, and 28 of the study. Blood glucose was measured by a one-touch electronic glucometer using glucose test strips. After 28 days, the rats were sacrificed, and blood samples were collected. Serum was examined for total cholesterol, high-density lipoprotein (HDL), triglycerides (TG), bilirubin, glutamate pyruvate transaminase (SGPT), glutamate oxaloacetate transaminase (SGOT), alkaline phosphatase (ALP), creatinine, and urea by commercially available kits (Erba Diagnostics, Mannheim, Germany).

### 2.12. Statistical Analysis

All assays were carried out in triplicate. The results were expressed as mean ± standard deviation (SD). The plots were prepared using Graphpad Prism software 5.01 version (GraphPad Software, San Diego, CA, USA). One-way analysis of variance (ANOVA) was used for statistical analysis. A *p-*value of less than 0.05 was considered statistically significant. 

## 3. Results

### 3.1. Phytochemical Analysis and Quantification of Total Phenolic and Flavonoid Contents

Phytochemical screening of *C. lanceolatus* methanolic and aqueous extracts revealed the presence of flavonoids, tannins, cardiac glycosides, terpenoids, reducing sugars, saponins, and phlobatannins (Table 1). Reducing sugars were present only in methanolic extract, while anthraquinones were absent in both the extracts. The total phenolic content was appreciably high (*p* < 0.001) in the methanolic extract as compared to aqueous extract. However, not much difference was observed in both the extracts with respect to flavonoid content (Table 1). 

### 3.2. DPPH Free Radical Scavenging Activity

The free radical scavenging activity of both the extracts at various concentrations (40–100 µg/mL) was assessed by measuring the extent of degree of discoloration of DPPH solution, which exhibits the scavenging ability of the extracts, and the results are presented in Figure 2. Considerable scavenging activity was observed for the methanolic (IC50 31.31 µg/mL) as well as an aqueous (IC50 50.89 µg/mL) extract in a concentration-dependent fashion. However, the activity increment was lower in the aqueous extract at all tested concentrations (40, 60, 80, and 100 µg/mL). At each concentration, the methanolic extract showed activity comparable to the standard antioxidant BHA (IC50 18.05 µg/mL). A statistically significant (*p* < 0.05) difference in DPPH free radical scavenging activity was observed between the aqueous extract and BHA. 

### 3.3. Reducing Power of the Extract

The reducing ability of *C. lanceolatus* extracts was measured in a concentration range of 100–1000 µg/mL. As depicted in Figure 3, a concentration-dependent reducing power was observed in both the extracts. The methanolic extract demonstrated a comparatively superior reducing activity to aqueous extract at all test concentrations. At 200 and 400 µg/mL concentrations, about a 2.5-fold higher reducing power was observed for methanolic extract compared to aqueous extract. With a further increase in concentration (600, 800 and 1000 µg/mL), the difference rose to about 3-fold, suggesting the appreciable reducing ability of the methanolic extract. The reducing activity of the methanolic extract was comparable to that of standard vitamin C at each concentration, whereas a significant (*p* < 0.01) decrease in the activity of the aqueous extract was observed compared to vitamin C.

### 3.4. Metal Ion Chelating Activity 

The ferrous ion chelating activity of *C. lanceolatus* aqueous and methanolic extracts was measured at various concentrations (100–400 µg/mL) and the results are presented in Figure 4. A concentration-dependent increase in chelating activity was found in both the extracts. The aqueous extract exhibited superior chelating activity to the methanolic extract at all test concentrations. At 100 µg/mL, the chelation ability of the aqueous extract (16%) was 2-fold higher compared to the methanolic extract (8.45%). At a concentration of 200 µg/mL, the chelation activities of the aqueous and methanolic extracts were 25.5% and 14.74%, respectively. With a successive increase in concentration up to 400 µg/mL, a further increment in chelating activity up to 42% was observed. However, the chelating activity of propyl gallate was considerably (*p* < 0.001) superior to both the extracts.

### 3.5. Effects of C. lanceolatus Extract on Animal Bodyweight and Blood Glucose Level

As illustrated in Figure 5A, a continuous increase in the bodyweight of the rats was observed in the normal control during a period of 28 days, while a decrease in body weight was observed in diabetic rats. However, the overall difference between the body weight of the normal control and diabetic rats was 75 g after 28 days (*p* < 0.01). At the same time, an increase (*p* < 0.05) in body weight was observed in antidiabetic drug (glibenclamide) treated-diabetic rats as compared to group 2 diabetic rats. Oral supplement of test extracts (200 or 400 mg/kg) displayed weight gain (*p* < 0.05) in comparison to diabetic control rats. However, overall weight gain in glibenclamide-treated rats was superior to that of the extract treatment group. No noticeable difference in the bodyweight of extract control and normal control was observed at any time point. Blood glucose in diabetic rats gradually increased from 280 mg/dL (day 1) to 370 mg/dL (day 28) while it was almost constant near 100 mg/dL in the normal control during the study period (Figure 5B). The difference in the blood glucose level between normal and diabetic rats was statistically significant (*p* < 0.001) after 28 days. No considerable diminution in sugar level was observed up to day 7 in glibenclamide-treated diabetic rats as compared to diabetic rats. However, a rapid decline in glucose level of glibenclamide treated rats was found after day 14 or day 28, indicating that a significant (*p* < 0.001) reduction in blood glucose. *C. lanceolatus* methanolic extract (200 or 400 mg/kg) administration in alloxan-induced diabetic rats for 28 days led to a significant (*p* < 0.05 or 0.01) decrease in blood glucose. The extract at 400 mg/kg exhibited a comparatively better hypoglycemic effect than the extract at 200 mg/kg. No alteration in blood sugar level was observed in the extract control as compared to normal rats.

### 3.6. Effect of C. lanceolatus Extract on Serum Total Cholesterol and Serum HDL

Diabetes is also associated with altered lipid profiles. A noteworthy increase in serum total cholesterol (*p* < 0.01) with a decrease in HDL cholesterol (*p* < 0.001) was observed in diabetic rats as compared to that of normal control (Figure 6A,B, respectively). Glibenclamide-treated hyperglycemic rats reveal a decline in total cholesterol (*p* < 0.01) with a marginal and insignificant increase in HDL cholesterol compared to diabetic rats. A methanolic extract (200 or 400 mg/kg) treatment of diabetic rats showed a decrease in total cholesterol (*p* < 0.05) and a slight increase (insignificant) in HDL levels compared to diabetic control. Insignificant changes were recorded in the lipid profiles in extract control rats compared to normal control.

### 3.7. Effect of C. lanceolatus Extract on Serum Triglyceride and Bilirubin

The effects of the extract on serum triglyceride and bilirubin in control and different groups are shown in Figure 7A,B, respectively. Diabetic rats showed raised levels of triglyceride (*p* < 0.001) and total bilirubin (*p* < 0.001) compared to the normal control. Glibenclamide treatment diminished bilirubin (*p* < 0.001) and triglyceride (*p* < 0.001) levels as compared to diabetic control. The plant extract (200 mg/kg) exhibited a considerable (*p* < 0.05) decline in triglyceride but an insignificant reduction in bilirubin. However, methanolic extract (400 mg/kg) treatment accounted for a notable decline in triglyceride (*p* < 0.01) and bilirubin (*p* < 0.05) levels. The result of the extract treatment alone was non-significant compared to that of normal animals.

### 3.8. Effect of C. lanceolatus Extract on Serum Enzymes

Hyperglycaemic rats exhibited significantly (*p* < 0.001) elevated activity of SGOT (Figure 8A), SGPT (Figure 8B), and ALP (Figure 8C) compared to the normal control. Glibenclamide treatment caused a reduction in the enzyme activities of ALP (*p* < 0.001), SGPT (*p* < 0.01), and SGOT (*p* < 0.001) compared to diabetic rats. The diabetic rats fed with *C. lanceolatus* methanolic extract at 200 mg/kg caused a considerable (*p* < 0.01) decline in the activity of SGOT and ALP compared to the diabetic control. Interestingly, the *C. lanceolatus* extract at 400 mg/kg appreciably (*p* < 0.05 or 0.001) reduced all enzyme activities compared to the diabetic control. No considerable variation was observed between extract control and normal control.

### 3.9. Effect of C. lanceolatus Extract on Serum Creatinine and Urea

Treatment with alloxan caused significant (*p* < 0.001) elevation in creatinine level (Figure 9A) and blood urea nitrogen (Figure 9B) compared to normal rats. Glibenclamide treatment accounted for a decrease in creatinine (*p* < 0.01) and blood urea level (*p* < 0.001) in comparison with the diabetic control. In diabetic rats fed with the methanolic extract (200 mg/kg), the effect of treatment on creatinine level was non-significant, while the higher dose of the extract (400 mg/kg) caused a considerable (*p* < 0.05) decline in creatinine level. The extract treatment (200 or 400 mg/kg) in diabetic rats caused a noteworthy (*p* < 0.01) reduction in blood urea nitrogen. There were no differences in creatinine and blood urea nitrogen levels between extract control and normal control animals.

## 4. Discussion

Oxidative stress represents a state of imbalance between free radical production and the response of the host endogenous antioxidant defence system. Several reports have indicated that DM is associated with increased free radical generation and a decrease in antioxidant status because hyperglycemia-mediated increase in protein glycosylation results in overproduction of free radicals [26]. Plant-based natural products have been used as therapeutic agents since time immemorial. They have the ability to alter human metabolism in a favorable way for the prevention of chronic and degenerative diseases [27,28]. In the present study, the antioxidant and anti-diabetic activities of *C. lanceolatus* were studied due to its traditional uses in the treatment of few diseases in various tribal cultures. Therefore, the study was performed to provide a scientific basis for its claimed therapeutic applications. Phenolics are the group of natural compounds reported to exhibit various beneficial effects, including their antioxidant properties. The stem of *C. lanceolatus* is rich in phenolics and flavonoid compounds [29,30]. The results reveal that methanol extract has higher content of phenolics as compared to the aqueous extract. This could probably be due to the higher polarity and better solubility of phenolic components present in plant materials in methanol [30]. The antioxidant activity of phenolics is due to their free radical scavenging ability conferred by their hydroxyl groups. Besides this, phenolic compounds effectively modulate polyol enzymes involved in diabetes-associated complications. Likewise, flavonoids possess significant scavenging activity against most of the oxidants, such as free radicals and singlet oxygen, involved in various disorders, including diabetes [31].

The DPPH assay revealed that methanolic and aqueous extracts of *C. lanceolatus* stem had appreciable radical scavenging ability in a concentration-dependent manner. More than 80% activity was recorded with the methanolic extract at a concentration of 100 µg/mL. The aqueous extract was comparatively less active. DPPH assay is mostly used as a reliable, quick, and reproducible method to ascertain the in vitro antioxidant activity of plant extracts. DPPH is a stable nitrogen-centered free radical and its color changes from violet to yellow upon reduction [32]. The DPPH scavenging activity of *C. lanceolatus* stem extracts can be correlated with number of available hydroxyl groups present in the phytocomponents of the stem extracts. It was further substantiated by the experimental data obtained during the reducing power assay of extracts at test concentrations. A continuous increase in reducing potential of the methanolic extract was observed with increasing concentration, suggesting its ability of electron-donation and thereby reducing Fe^3+^ into Fe^2+^ complex in vitro. The activity could be correlated with the presence of higher content of phenolics in methanolic extract than aqueous extract. Phenolic compounds act as reductones by contributing electrons and thereby reacting with free radicals to convert them to harmless and more stable products [33].

The iron, a transition metal, has the capability to donate single electrons which paves the way for generation and propagation of various radical reactions. Hence, the removal of transition metals by chelation is another strategy to diminish free radical generation in the system. During metal ion chelating assay, the methanolic extract showed a lower chelating ability than aqueous extract which showed about 35% activity at the highest test concentration. However, the activity of the standard chelating agent was higher than both the extracts. It has been reported that phenolics and flavonoids mediate the chelation of iron through the formation of sigma bonds with metals. They decrease the redox potential and thus stabilize the oxidized state of the metal ion [34].

Since the methanolic extract displayed comparatively better in vitro antioxidant efficacy, it was further assessed for antihyperglycemic activity in alloxan-induced diabetic rats. The alloxan action in the pancreas is initiated by its rapid uptake by pancreatic β-cells causing toxicity and damage by producing free radicals that result in decreased insulin secretion. It has been suggested that the action of glibenclamide in a diabetic condition is mediated by stimulating insulin production from the remaining pancreatic β-cells and causing a blood glucose lowering effect. Several late complications of diabetes are known to be caused due to underlying oxidative stress subsequent to persistent hyperglycemia. Earlier studies have tested the efficacy of antioxidant food supplements as a strategy for preventing/managing the development of diabetic complications [34]. The repetitive dosing of *C. lanceolatus* methanolic extract over a period of 4 week resulted in the maintenance of body weight with a decreased blood glucose level in diabetic rats. The antioxidant and free radical scavenging effects of *C. lanceolatus* might be implicated in their hypoglycemic and antidiabetic activities. The present study supported these facts, as the extract had the ability to decrease the glucose levels in diabetic rats. 

Serum lipid profiling revealed that the alloxan administration resulted in the hyperlipidemic condition characterized by significant enhancement in the level of serum triglyceride and total cholesterol, as compared to normal rats. A significant reduction in the serum HDL cholesterol level was also observed in alloxan-treated diabetic rats. These results are in agreement with the previously reported alloxan-induced hyperlipidemic condition in diabetic rats [6,35]. The extract caused a decrease in serum total cholesterol and triglyceride, but the effect on HDL level was insignificant in diabetic rats. Hence, this action of extract suggests that hypolipidemic action is not simply secondary to its blood glucose lowering effect. Hyperglycemia has been related to the elevated serum level of total cholesterol, triglyceride, urea, and creatinine. Reports indicate that flavonoids are well known for their activities as antioxidant and hypocholesterolemic agent. In addition to this, flavonoids are well established as a radical scavenger. This attribute has been associated with lowering of hyperglycemia as well as hyperlipidemia [35,36]. The hypolipidemic action of the methanolic extract could be an additional advantage of the use of the extract in the management of diabetes. The mechanism of action of the extract is not clearly known, but it could be possible that it effects its hypoglycemic action by upsurge of the peripheral utilization of glucose. 

The inadequacy of insulin enhances the levels of triglyceride and cholesterol by the inactivation of rate-limiting enzyme lipoprotein lipase and HMG-CoA reductase, respectively. It has been suggested that the mode of action of glibenclamide in hyperglycemic condition involves lowering blood glucose via stimulating insulin production from the existing β-cells of the pancreas [36]. Phytochemicals stimulate adiponectin secretion that causes the probable enhanced activity of the peroxisome proliferator-activated receptor-γ that leads to an increased glucose uptake. It also activates adenosine monophosphate-activated protein kinase followed by GLUT4 translocation to the plasma membrane of muscle cells and enhances glucose homeostasis in type 2 diabetes [37,38].

The level of bilirubin increased in diabetic rats appreciably in comparison to normal rats. Glibenclamide or *C. lanceolatus* stem extract (400 mg/kg) treatment of diabetic rats accounted for a reduction in serum bilirubin. For quite a long-time, bilirubin was accepted to be just a waste product of heme catabolism and possibly toxic compound at worst. However, recent experimental findings have indicated that a somewhat raised serum concentration of bilirubin has a strong correlation with the lesser occurrence of oxidative stress-mediated disorders. Serum bilirubin has been reliably demonstrated to be negatively correlated with diabetes mellitus, arterial hypertension, metabolic syndrome, obesity, and cardiovascular disease [39]. In a cross-sectional study, Japanese investigators reported that in comparison to subjects with lower bilirubin, those with the highest bilirubin levels had a four times lower prevalence of diabetic retinopathy [40].

It is well known that a diabetic state is associated with hepatocellular damage and leakages of liver enzymes, leading to an increase in SGOT, SGPT, and ALP. The *C. lanceolatus* methanolic extract used in this study contains numerous phytochemicals and especially flavonoids that are associated with reversing such changes [41,42]. *C. lanceolatus* extract was found to reduce the SGPT and ALP levels on day 28, similar to glibenclamide. ALP, a cytoplasmic enzyme, is released into the circulation following cellular damage. Besides this, the soluble enzymes, such as SGOT and SGPT, are released when the injury involves organelles, such as mitochondria. Thus, the alloxan-induced hepatic damage resulted in injury at cell membrane as well as organelle membrane level. It has been reported that the elevated activities of these enzymes are indicative of cellular leakage and the loss of the functional integrity of the cell membranes. It is well known that a diabetic state is associated with hepatocellular damage and leakages of liver enzymes, leading to an increase in SGOT, SGPT, and ALP, implicating on the hepatotoxic impact of alloxan [43,44]. In the present study, alloxan administration resulted into hepatic damage, followed by the elevated level of serum level of hepatic enzymes SGOT, SGPT, and ALP. 

Experimentally, the *C. lanceolatus* methanolic extract showed antioxidant activity in vitro and thereby inhibited free radical-mediated toxicity. Hence, it was responsible for producing hypoglycemic action by inhibiting ROS-induced β-cell damage at the pancreatic level along with improving hepatic functions, as indicated by liver function markers. Thus, the antidiabetic activity of *C. lanceolatus* stem extract finds a direct correlation with the antioxidant activity of the extract. 

Creatinine is the principal waste product of creatine metabolism. In the kidney, it is filtered by the glomerulus and effectively discharged by the tubules. In addition, free creatinine shows up in the blood serum. Urea is the main waste products of protein catabolism. Liver is the site of urea synthesis from amino acid metabolism. The creatinine is a waste product produced by muscle from the breakdown of creatine, which is also synthesized in the liver. High serum creatinine level is also a reliable marker of muscle tissue damage. Renal function tests help to determine if the kidney is performing their task adequately [42]. Experimental findings have indicated that the elevated blood urea nitrogen level causes insulin resistance and diminishes insulin secretion. However, in humans, the correlation between a higher blood urea nitrogen level and increased risk of diabetes is not well known. Higher serum creatinine and urea levels, coupled with a decreased urinary excretion of creatinine, are indicators of diabetic nephropathy. The reversal of these effects was observed in diabetic rats treated with *C. lanceolatus* extract. A higher concentration of extract (400 mg/kg) accounted for a significant decline in creatinine and urea.

*C. lanceolatus* has been utilized in folk medicine and the pharmacological activities of leaves are widely studied. The phenolic and flavonoid contents present in different extracts of *C. lanceolatus* leaves have been shown to exhibit free radical scavenging activity. Their occurrence in stem extract could be ascribed to the antioxidant ability of phytoconstituents. Phytochemicals present in leaf ethyl acetate and methanolic extracts, such as 4-fluoro-2-trifluoromethylbenzoic acid, neopentyl ester; fumaric acid, di(pent-4-en-2-yl) ester; 2,3-dihydro-3,5-dihydroxy-6-methyl-4H-pyran-4-one and 2-furancarboxaldehyde,5-(hydroxymethyl), have been shown to possess anticancer and antioxidant activities [12]. In addition, the presence of alkanol, flavones, and triterpenoids might be responsible for various biological activities [45,46]. *C. lanceolatus* stem extracts have not been fully characterized so far. To best of our knowledge, this is the first report on the antihyperglycemic activity of *C. lanceolatus* stem methanolic extract. Further investigations are required to identify the specific phytochemicals responsible for the antihyperglycemic activity of the extract. 

## 5. Conclusions

The results of our study demonstrate for the first time that methanolic and aqueous extracts of *C. lanceolatus* stem exhibit substantial antioxidant activity. The methanolic extract of *C. lanceolatus* has considerable hypoglycemic and antidiabetic activities. In addition, the methanolic extract improves the associated complications of DM, as characterized by several parameters tested, such as body weight and lipid profiles along with the serum markers of hepatic and renal functions, namely SGPT, SGOT, ALP, bilirubin, creatinine, and blood urea nitrogen. Additional studies are warranted to understand the full potential of the methanolic extract of *C. lanceolatus* stem for the therapeutic management of DM.

## Figures and Tables

**Figure 1 medicines-07-00011-f001:**
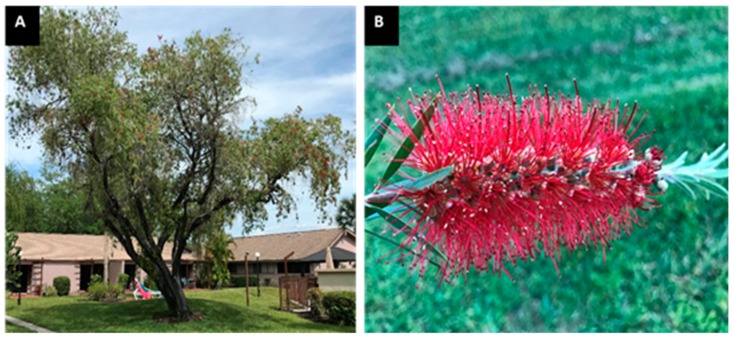
Images of *C. lanceolatus* whole tree (**A**) and flowers (**B**).

**Figure 2 medicines-07-00011-f002:**
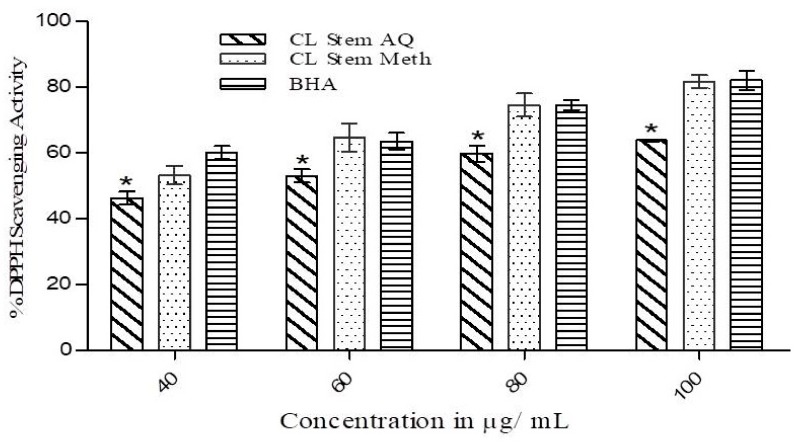
Free radical scavenging activity of methanolic (Meth) and aqueous (AQ) extracts of *C. lanceolatus* stem. Radical scavenging activity was determined at various concentrations of both extracts (40–100 µg/mL). Butylated hydroxyl anisole (BHA) was used as a control. The results are expressed as a mean ± SD of three replicates. * *p* ˂ 0.05 compared to BHA.

**Figure 3 medicines-07-00011-f003:**
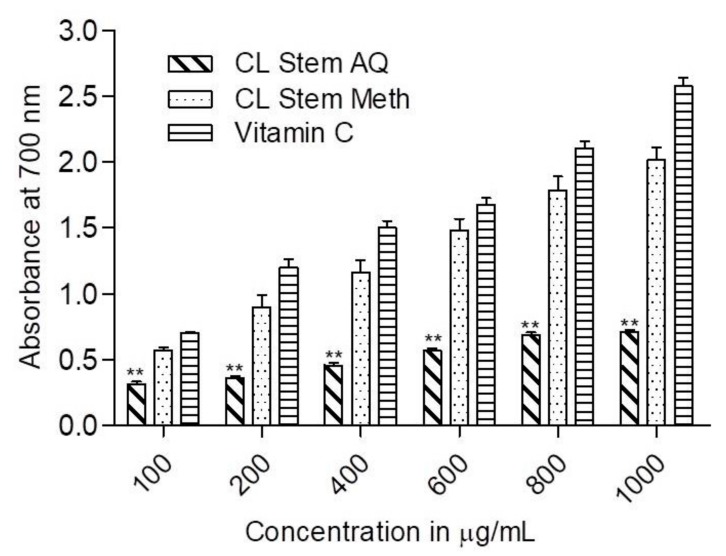
Reducing power of *C. lanceolatus* stem methanolic (Meth) and aqueous (AQ) extracts. The reducing power was measured at different extract concentrations (200–1000 µg. Vitamin C was used for comparison. The results are expressed as a mean ± SD of three replicates. ** *p* < 0.01 compared to vitamin C.

**Figure 4 medicines-07-00011-f004:**
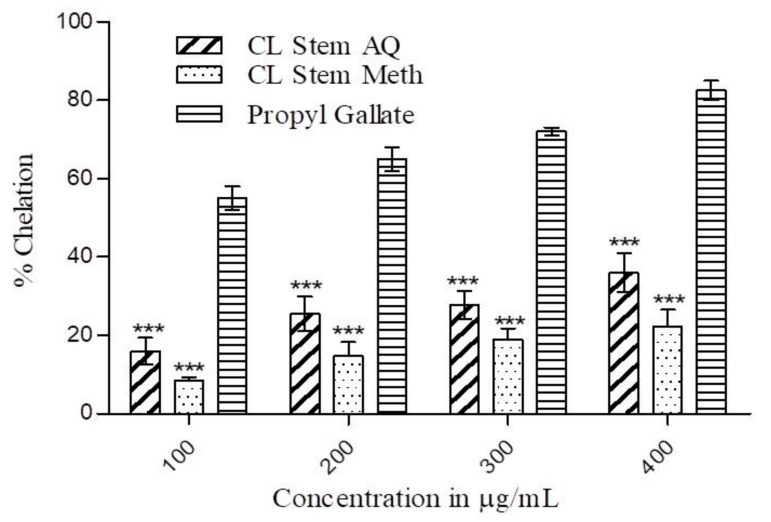
Metal ion chelation activity of *C. lanceolatus* stem methanolic (Meth) and aqueous (AQ) extracts. The chelating activity of extracts was measured in concentration range of 100–400 µg/mL. Propyl gallate was used for comparison. The results are expressed as a mean ± SD of three replicates. *** *p* ˂ 0.001 compared to propyl gallate.

**Figure 5 medicines-07-00011-f005:**
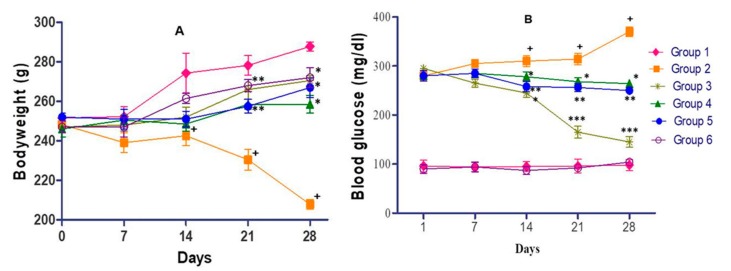
Effect of *C. lanceolatus* methanolic extract on body weight (**A**) and blood glucose level (**B**) of rats. The results are expressed as a mean ± SD (*n* = 5). Group 1: Normal control; Group 2: Diabetic control; Group 3: Diabetic rats treated with glibenclamide; Group 4: Diabetic rats fed with *C. lanceolatus* extract (200 mg/kg); Group 5: Diabetic rats fed with *C. lanceolatus* extract (400 mg/kg); Group 6: *C. lanceolatus* extract (400 mg/kg) control. ^+^
*p* < 0.01 compared to normal control. * *p* < 0.05, ** *p* < 0.01 and *** *p* < 0.001 compared to the diabetic control.

**Figure 6 medicines-07-00011-f006:**
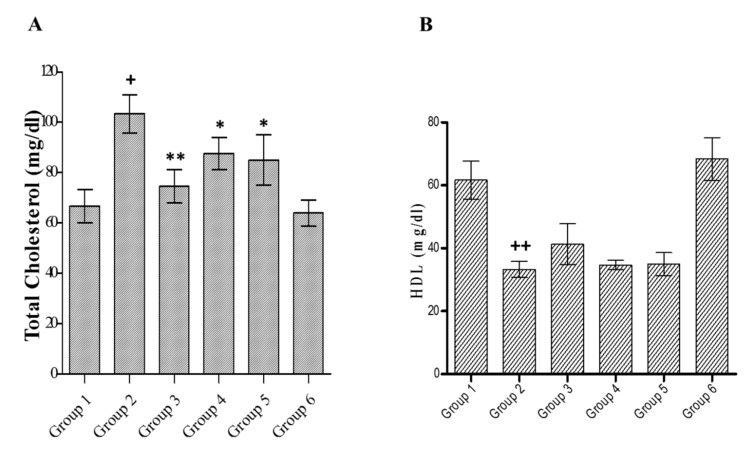
Effect of *C. lanceolatus* methanolic extract on serum total cholesterol (**A**) and high-density lipoprotein (HDL) (**B**) levels. The results are expressed as a mean ± SD (*n* = 5). Group 1: Normal control; Group 2: Diabetic control; Group 3: Diabetic rats treated by glibenclamide; Group 4: Diabetic rats fed with *C. lanceolatus* extract (200 mg/kg); Group 5: Diabetic rats fed with *C. lanceolatus* extract (400 mg/kg); Group 6: *C. lanceolatus* extract (400 mg/kg) control. ^+^
*p* < 0.01 and ^++^
*p* < 0.001 compared to normal control. * *p* < 0.05 and ** *p* < 0.01 compared to the diabetic control.

**Figure 7 medicines-07-00011-f007:**
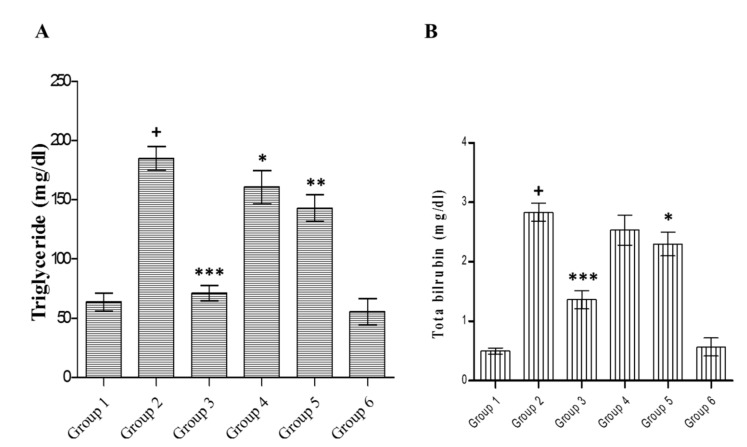
Effect of *C. lanceolatus* methanolic extract on serum triglyceride (**A**) and bilirubin (**B**) levels. The results are expressed as a mean ± SD (*n* = 5). Group 1: Normal control; Group 2: Diabetic control; Group 3: Diabetic rats treated with glibenclamide; Group 4: Diabetic rats fed with *C. lanceolatus* extract (200 mg/kg); Group 5: Diabetic rats fed with *C. lanceolatus* extract (400 mg/kg); Group 6: *C. lanceolatus* extract (400 mg/kg) control. ^+^
*p* < 0.001 compared to normal control. * *p* < 0.05, ** *p* < 0.01 and *** *p* < 0.001 compared to the diabetic control.

**Figure 8 medicines-07-00011-f008:**
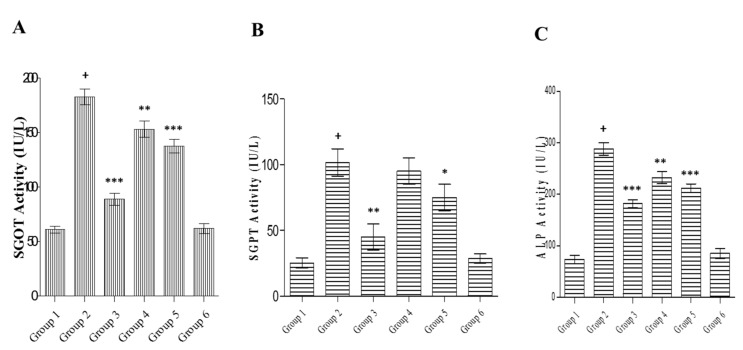
Effect of *C. lanceolatus* methanolic extract on serum glutamate oxaloacetate transaminase (SGOT) (**A**), glutamate pyruvate transaminase (SGPT) (**B**) and alkaline phosphatase (ALP) (**C**) levels. The results are expressed as a mean ± SD (*n* = 5). Group 1: Normal control; Group 2: Diabetic control; Group 3: Diabetic rats treated by glibenclamide; Group 4: Diabetic rats fed with *C. lanceolatus* extract (200 mg/kg); Group 5: Diabetic rats fed with *C. lanceolatus* extract (400 mg/kg); Group 6: *C. lanceolatus* extract (400 mg/kg) control. ^+^
*p* < 0.001 compared to normal control. * *p* < 0.05, ** *p* < 0.01 and *** *p* < 0.001 compared to the diabetic control.

**Figure 9 medicines-07-00011-f009:**
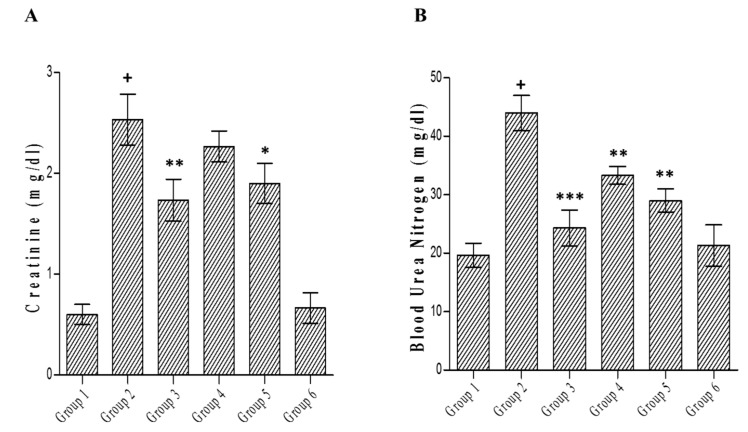
Effect of *C. lanceolatus* methanolic extract on serum creatinine (**A**) and blood urea nitrogen (**B**) levels. The results are expressed as a mean ± SD (*n* = 5). Group 1: Normal control; Group 2: Diabetic control; Group 3: Diabetic rats treated with glibenclamide; Group 4: Diabetic rats fed with *C. lanceolatus* extract (200 mg/kg); Group 5: Diabetic rats fed with *C. lanceolatus* extract (400 mg/kg); Group 6: *C. lanceolatus* methanolic extract (400 mg/kg) control. ^+^
*p* < 0.001 compared to normal control. * *p* < 0.05, ** *p* < 0.01 and *** *p* < 0.001 compared to the diabetic control.

**Table 1 medicines-07-00011-t001:** Qualitative and quantitative assessment of *C. lanceolatus* stem extracts.

Phytochemicals	Extract		Quantitative Analysis			
	Meth	AQ				
Tannins	+	+	**Total Phenols** **(µgPGE/mg)**		**Total Flavonoids** **(µgQE/mg)**	
Flavonoids	+	+				
Terpenoids	+	+	**AQ**	**Meth**	**AQ**	**Meth**
Cardiac glycosides	+	+				
Anthraquinones	-	-	72.27 ± 0.55	227.96 ± 0.17 *	10.13 ± 0.27	10.49 ± 0.42
Reducing sugars	+	-				
Phlobatannins	+	+				
Saponins	+	+				

Meth, Methanolic extract; AQ, Aqueous extract; PGE, propyl gallate equivalent; QE, quercetin equivalent. * *p* < 0.001 compared to aqueous extract. Results are expressed as a mean ± SD (*n* = 3).

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
