# Peer review of "The Antioxidant and Antihyperglycemic Activities of Bottlebrush Plant (Callistemon lanceolatus) Stem Extracts"

_medicines, 2020, doi:10.3390/medicines7030011_

Round 1

Reviewer 1 Report

This is a well described study in the antioxidant and antihyperglycemic properties of bottlebrush stem extracts. Various in vitro methods were employed along with animal study using alloxan injection in rodent as a diabetes model. However, there are several points to concern as follows:

Introduction:

  • Please re-check the correction of references on each paragraph. For example, line 51-53, there is no information in ref# 5. Line 57-64, ref# 3 and 7 are in vitro studies that appear unrelated to the sentences. Also line 71-75, it seems no related information in ref#10.
  • Plagiarism detection on line 54-57 “According to the World Health Organization (WHO) report, the global population is in the middle of a diabetes epidemic. The people in Southeast Asia and the Western Pacific are under greater risk, and the majority of them have type-2 diabetes mellitus (T2DM)”: the same sentences were written in ref#5!

Materials and Methods:

  • 12—if post hoc analysis was utilized, please identify.

Results:

  • 1—It would be useful and interesting to add a table of phytochemical components and results together with total phenols and flavonoids.
  • Figure 5 and 6—the authors might consider using similar symbols for the same group of treatment.

Discussion:

  • Line 369-370—according to Figure 2, it seems that the extract at concentration 80 mg/ml does not reach 90% activity.
  • Line 408—from Figure 7, the increase of HDL from the extract is insignificant and appear similar to diabetic control group.
  • Line 426 and 465—as shown in Figure 8 and 10, lanceolatus stem extract at 200 mg/kg has no significant difference on bilirubin and creatinine levels.
  • Line 438—considering the results from Figure 9, SGOT levels from the extract are more similar to SGPT in which there is no statistical difference at concentration 200 mg/kg.

Author Response

The authors of this manuscript express their sincere thanks to the and reviewer for the critical assessment of our work. The authors have acted upon the recommendations of the reviewers as well as the managing editor which has resulted in a significant enhancement of the quality of this manuscript. All modifications incorporated in the manuscript are highlighted using red color font. A “point-by-point” response to the reviewers’ comments is outlined below.

General comments:

This is a well described study in the antioxidant and antihyperglycemic properties of bottlebrush stem extracts. Various in vitro methods were employed along with animal study using alloxan injection in rodent as a diabetes model. However, there are several points to concern as follows:

Response:

We are deeply encouraged by the generous comments of the reviewers. We have addressed the specific concerns and revised our manuscript as described below.

Specific comments:

Comment on Introduction:

  • Please re-check the correction of references on each paragraph. For example, line 51-53, there is no information in ref# 5. Line 57-64, ref# 3 and 7 are in vitro studies that appear unrelated to the sentences. Also line 71-75, it seems no related information in ref#10.
  • Plagiarism detection on line 54-57 “According to the World Health Organization (WHO) report, the global population is in the middle of a diabetes epidemic. The people in Southeast Asia and the Western Pacific are under greater risk, and the majority of them have type-2 diabetes mellitus (T2DM)”: the same sentences were written in ref#5!

Response:

  1. We thank reviewer for highlighting a very important point. We have rectified the errors as indicated below:
  • For the lines 51-53: (now lines 50-52), the correct reference 4 has been incorporated in place of reference 5.
  • For lines 57-64: reference 3 has been deleted from line 61 and reference 7 has been replaced with a new relevant reference (Dos Santos et al., 2019) under the reference section.
  • Line 71-75: reference 10 has been replaced with a new relevant reference (Marzouk et al., 2008) under the reference section.
  1. Plagiarism issues on line 54-57: Plagiarism issues have been resolved in the manuscript as a whole as well as for sentence indicated (page 2, lines 54-56).

Comment on Materials and Methods:

 2.12—if post hoc analysis was utilized, please identify.

 Response:

  • Post hoc analysis was not utilized. We have used one-way ANOVA only as indicated on page 5, line 195.

 Comment on Results:

  •  1—It would be useful and interesting to add a table of phytochemical components and results together with total phenols and flavonoids.
  • Figure 5 and 6—the authors might consider using similar symbols for the same group of treatment.

 Response:

  • We sincerely thank the reviewer to this excellent suggestion. Table 1 has been modified to include phytochemical components together with results on total phenols and flavonoids. The title of Table 1 has been changed accordingly to “Qualitative and quantitative assessment of lanceolatus stem extracts” (page 5).
  • Figure 5 and 6 have been combined as Figure 5 (A and B), and similar symbols have been used to depict the same group of treatment.

 Comment on Discussion:

  •  ine 369-370—according to Figure 2, it seems that the extract at concentration 80 mg/ml does not reach 90% activity.
  • Line 408—from Figure 7, the increase of HDL from the extract is insignificant and appear similar to diabetic control group.
  • Line 426 and 465—as shown in Figure 8 and 10, C. lanceolatus stem extract at 200 mg/kg has no significant difference on bilirubin and creatinine levels.
  • Line 438—considering the results from Figure 9, SGOT levels from the extract are more similar to SGPT in which there is no statistical difference at concentration 200 mg/kg.

 Response:

  •  Line 369-370 and Figure 2: We admire the reviewer for his/her careful observation. We regret our oversight and have modified the statement as: “More than 80% activity was recorded with the methanolic extract at a concentration of 100 µg/ml” (page 11, lines 368 and 369).

  • Line 408 and Figure 7: the increase of HDL from the extract is insignificant and appear similar to diabetic control group. We thank the reviewer for this keen observation. The error has been corrected and the sentence has been modified as: “The extract caused a decrease in serum total cholesterol and triglyceride, but the effect on HDL level was insignificant in diabetic rats.” (page 12, lines 406-408).

  • Line 426 and 465 and Figure 8 and 10: The reviewer is correct that lanceolatus stem extract at 200 mg/kg has no significant effect on bilirubin level. The extract at higher concentration (400 mg/kg body weight) exhibited significant reduction in bilirubin. The necessary changes have been incorporated in the revised text (page 12, line 426 and page 13, line 427).  Similar correction has been made regarding discussion on creatinine (page 13, lines 464 and 465).

  • Line 438 and Figure 9 (now Figure 8): The SGOT activity in group 4 at a concentration of 200 mg/kg is significant in comparison with diabetic control. However, SGPT activity in group 4 rats is not significant at 200 mg/kg in comparison with diabetic control.

Additionally,

  1. The reference list has been modified as we have added several new references. Special attention is given to conform to the order of references and bibliographic style of the journal.
  2. The entire manuscript has been thoroughly checked and edited to ensure uniform style, organization, and quality for the English language.

On behalf of my co-authors, we once again express our sincere thanks to the erudite reviewer for the valuable suggestions and constructive input to improve the quality of our manuscript.

Reviewer 2 Report

The study presents in vitro studies of antioxidant activity of methanolic and aqueous extracts from Callistemon lanceolatus stem and their in vivo antihyperglycemic activities.  

The paper presents novelty considering that in the literature were not reported before studies on the antihyperglycemic activities of  Callistemon lanceolatus stem extracts and very few reports regarding the in vivo antioxidant activity however the article requires some minor revisions that need to be addressed prior to the publication.

1) Page 1-line 2

    I would suggest the title to be changed into: “The antioxidant and antihyperglycemic activities of Bottlebrush plant (Callistemon lanceolatus) stem extracts”.

2) Page 2-Introduction (line 86)

I would change Figure 1. legend for “Images of C. lanceolatus whole tree (A) and flowers (B), respectively”.

3) Page 3- 2.3-2.5

    I would consider that in the Materials and methods section to reunite subsections 2.3-2.5 into one, similarly to the subsection from the Results section.

4) Page 3-Determination of total phenolics (line 122)

I would suggest changing the units from mg/g to µg/mg for a better consistency considering that further (Table 1) are mentioned in µg/mg.

5) Page 4-Free radical scavenging assay (line 134), Metal ion chelating activity (line 155)

I would suggest explaining what is considered control in each case (for example, for DPPH assay it could be added Control- DPPH free radical solution).

6) Page 5- Phytochemical analysis and quantification of total phenolic and flavonoid contents

-line 201

I would pay attention to adding italics for the Latin name of the plant “C. lanceolatus”.

-line 204-206

Also, I would recommend adding an explanation for an increased polyphenolic content in the methanolic extract or at least support the obtained values with literature data from which methanol is leading to a higher total polyphenols content (not necessary for the same plant extracts if not applicable).

7) Page 6-DPPH radical scavenging activity (line 210), Metal ion chelating ability (line 234)

I think it would be best to change the title into “DPPH free radical scavenging activity” and also from the acquired data to calculate IC50% (concentration that is needed to inhibit 50% of the DPPH free radical) for extracts and BHA and compare the extracts using this concentration being significant for the antioxidant power of samples. Same IC 50% can be computed for Metal ion chelating ability also.

8) Page 8 – Figure 5 and Figure 6 (Line 274, 280)

I would think that Figure 5 and 6 could be turned into one figure (A, B) and would be best to use the same symbols for the same group data and also adding different colors because it is being difficult to see the data points for each group considering all are represented in black line. 

9) Page 12– Discussion (Line 375)

I recommend adding the word phytocomponents into the phrase “The reduction in the number of DPPH molecules can be correlated with the number of available hydroxyl groups present in C. lanceolatus stem extracts phytocomponents” considering that hydroxyl groups are part of extract components structure and not parts of an extract.

10) Page 13– Discussion (Line 450)

I would suggest not to affirm that you proved in vivo antioxidant activity just by making in vitro tests considering that in vivo antioxidant activity are a whole different subject and unless you also tested extract activity on cells (for example, by DCFDA assay) I would only leave “in vitro activity” mention in manuscript text.

Author Response

The authors of this manuscript express their sincere thanks to the reviewer for the critical assessment of our work. The authors have acted upon the recommendations of the reviewers as well as the managing editor which has resulted in a significant enhancement of the quality of this manuscript. All modifications incorporated in the manuscript are highlighted using red color font. A “point-by-point” response to the reviewers’ comments is outlined below.

General comments:

The study presents in vitro studies of antioxidant activity of methanolic and aqueous extracts from Callistemon lanceolatus stem and their in vivo antihyperglycemic activities.  

The paper presents novelty considering that in the literature were not reported before studies on the antihyperglycemic activities of Callistemon lanceolatus stem extracts and very few reports regarding the in vivo antioxidant activity however the article requires some minor revisions that need to be addressed prior to the publication.

Response:

We are indebted to the reviewer for his/her generous comments about the quality of our work. We have taken care of the minor revisions as suggested by the reviewer.

Specific comments:

Comment 1:

Page 1-line 2:  I would suggest the title to be changed into: “The antioxidant and antihyperglycemic activities of Bottlebrush plant (Callistemon lanceolatus) stem extracts”.

Response:

This is an excellent suggestion. The title has been changed as recommended by the reviewer.

Comment 2:

Page 2-Introduction (line 86): I would change Figure 1. legend for “Images of C. lanceolatus whole tree (A) and flowers (B), respectively”.

Response:

The legend to Figure 1 has been modified as suggested (page 2, line 85).

Comment 3:

Page 3- 2.3-2.5: I would consider that in the Materials and methods section to reunite subsections 2.3-2.5 into one, similarly to the subsection from the Results section.

Response: 

While we appreciate the comment, we sincerely believe that it would be extremely challenging to reunite subsections 2.3-2.5 because 2.3 deals with qualitative phytochemical screening of the extracts and 2.4 & 2.5 separately deal with the detailed processes of quantitative estimation of the flavonoids and phenolics.

Comment 4:

Page 3-Determination of total phenolics (line 122): I would suggest changing the units from mg/g to µg/mg for a better consistency considering that further (Table 1) are mentioned in µg/mg.

Response:

We agree with the reviewer. The necessary corrections have been incorporated in the text (page 3, lines 119 and 120).

Comment 5:

Page 4-Free radical scavenging assay (line 134), Metal ion chelating activity (line 155): I would suggest explaining what is considered control in each case (for example, for DPPH assay it could be added Control- DPPH free radical solution).

Response:

This is an excellent point. DPPH-free radical solution is considered as control in free radical scavenging assay while a ferrous chloride-ferrozine reaction mixture is used as control in metal ion chelating activity. The necessary changes have been incorporated in the text (page 3, line 128 and page 4, line 148, respectively).

Comment 6:

Page 5- Phytochemical analysis and quantification of total phenolic and flavonoid contents

-line 201

I would pay attention to adding italics for the Latin name of the plant “C. lanceolatus”.

-line 204-206

Also, I would recommend adding an explanation for an increased polyphenolic content in the methanolic extract or at least support the obtained values with literature data from which methanol is leading to a higher total polyphenols content (not necessary for the same plant extracts if not applicable).

Response:

We admire the reviewer for his/her watchful eyes. The Latin name of the plant has been italicized (page 5, line 199).

This is an excellent suggestion. The C. lanceolatus methanolic extract was found to have higher content of phenolics compared to the aqueous extract. This could probably be due to the higher polarity and better solubility of phenolic components present in plant materials in methanol. These lines have been incorporated in the discussion section (page 11, lines 360-362). The supportive reference (ref. no. 30) has been added in the bibliography section (Babbar, N.;  Oberoi, H.S.; Sandhu, S.K.; Bhargav, V.K. Influence of different solvents in extraction of phenolic compounds from vegetable residues and their evaluation as natural sources of antioxidants. J Food Sci Technol. 2014, 51, 2568–2575). 

Comment 7:

Page 6-DPPH radical scavenging activity (line 210), Metal ion chelating ability (line 234): I think it would be best to change the title into “DPPH free radical scavenging activity” and also from the acquired data to calculate IC50% (concentration that is needed to inhibit 50% of the DPPH free radical) for extracts and BHA and compare the extracts using this concentration being significant for the antioxidant power of samples. Same IC 50% can be computed for Metal ion chelating ability also.

Response:

We have changed the section title to “DPPH free radical scavenging activity” (page 6, line 208).

The IC50 values of the extracts for DPPH assay were found to be as follows: aqueous (IC50 50.89 µg/ml), methanolic (IC50 31.31 µg/ml) and BHA (IC50 18.05 µg/ml). All these have been incorporated in our revised manuscript (page 6, lines 212 and 215). However, the metal ion chelating activity (maximum chelation) observed with the extracts was 42%. Hence it is not relevant to extrapolate the IC50 values.

Comment 8:

Page 8 – Figure 5 and Figure 6 (Line 274, 280): I would think that Figure 5 and 6 could be turned into one figure (A, B) and would be best to use the same symbols for the same group data and also adding different colors because it is being difficult to see the data points for each group considering all are represented in black line. 

Response:

We are in absolute agreement with the reviewer. Accordingly, Figures 5 and 6 have been consolidated as Figure 5 (with panels A and B). We have also used different symbols and colors to depict treatments for various animal groups. We also have used uniform pattern for both the panels.

Comment 9:

Page 12 – Discussion (Line 375): I recommend adding the word photo components into the phrase “The reduction in the number of DPPH molecules can be correlated with the number of available hydroxyl groups present in C. lanceolatus stem extracts phytocomponents” considering that hydroxyl groups are part of extract components structure and not parts of an extract.

Response:

We have modified the sentence and added the word “phytocomponents” as suggested (page 11, lines 372-374).

Comment 10:

Page 13 – Discussion (Line 450): I would suggest not to affirm that you proved in vivo antioxidant activity just by making in vitro tests considering that in vivo antioxidant activity are a whole different subject and unless you also tested extract activity on cells (for example, by DCFDA assay) I would only leave “in vitro activity” mention in manuscript text.

Response:

We think the reviewer has made an excellent point. We have revised the sentence as suggested (page 13, lines 449 and 450).

Additionally,

  1. The reference list has been modified as we have added several new references. Special attention is given to conform to the order of references and bibliographic style of the journal.
  2. The entire manuscript has been thoroughly checked and edited to ensure uniform style, organization and quality for the English language.

On behalf of my co-authors, we once again express our sincere thanks to the erudite reviewer for the constructive input to improve the quality of our manuscript.